# Is Probiotic Supplementation Useful for the Management of Body Weight and Other Anthropometric Measures in Adults Affected by Overweight and Obesity with Metabolic Related Diseases? A Systematic Review and Meta-Analysis

**DOI:** 10.3390/nu13020666

**Published:** 2021-02-19

**Authors:** Simone Perna, Zahra Ilyas, Attilio Giacosa, Clara Gasparri, Gabriella Peroni, Milena Anna Faliva, Chiara Rigon, Maurizio Naso, Antonella Riva, Giovanna Petrangolini, Ali A. Redha, Mariangela Rondanelli

**Affiliations:** 1Department of Biology, College of Science, University of Bahrain, Sakhir Campus P. O. Box 32038, Sakhir, Bahrain; simoneperna@hotmail.it (S.P.); zahra.muhammadilyas@gmail.com (Z.I.); 2Gastroenterology and Digestive Endoscopy Unit, Centro Diagnostico Italiano, 20147 Milan, Italy; attilio.giacosa@gmail.com; 3Endocrinology and Nutrition Unit, Azienda di Servizi alla Persona ‘‘Istituto Santa Margherita’’, University of Pavia, 27100 Pavia, Italy; gabriella.peroni01@universitadipavia.it (G.P.); milena.faliva@gmail.com (M.A.F.); rigon.chiara2015@gmail.com (C.R.); mau.na.mn@gmail.com (M.N.); 4Research and Development Unit, Indena SpA, 20139 Milan, Italy; antonella.riva@indena.com (A.R); gio-vanna.petrangolini@indena.com (G.P.); 5Department of Chemistry, College of Science, University of Bahrain, Sakhir Campus P. O. Box 32038, Sakhir, Bahrain; ali96chem@gmail.com; 6Department of Public Health, Experimental and Forensic Medicine, University of Pavia, 27100 Pavia, Italy; mariangela.rondanelli@unipv.it; 7Istituto di Ricovero e Cura a Carattere Scientifico, Mondino Foundation, 27100 Pavia, Italy

**Keywords:** probiotics, weight loss, obesity, body weight

## Abstract

The aim of this systematic review and meta-analysis is to assess the effectiveness of probiotics in inducing body weight loss in patients with overweight or obesity with related metabolic diseases. The research was carried out on PubMed and Scopus, focusing on studies reporting the effect on anthropometric measures (weight, body mass Index (BMI), waist circumference (WC), and hip circumference (HC) after administration of various probiotic strains compared to placebo. Twenty randomized controlled trials, that included 1411 patients, were considered. The meta-analyzed mean differences (MD) for random effects showed no significant decrease in body weight after probiotic supplementation (−0.26 kg [−075, 0.23], *p* = 0.30), while a significant BMI decrease was found (−0.73 kg/m^2^ [−1.31, −0.16], *p* = 0.01). For WC and HC, the meta-analyzed MD for random effects showed a significant decrease (WC: −0.71 cm [−1.24; −0.19], *p* = 0.008 and HC: −0.73 cm [−1.16; −0.30], *p* = 0.0008). The risk of bias was also evaluated considering a high risk and a low risk according to PRISMA criteria. In conclusion, the results of this meta-analysis highlight a positive trend of probiotics supplementation on the amelioration of anthropometric measures of overweight and obese patients with related metabolic diseases. However, further research is needed before recommending the use of probiotics as a therapeutic strategy for these patients. The focus of the future research should be to evaluate the efficacy of different probiotic strains, the quantities to be administered, and the duration of the intervention.

## 1. Introduction

The role of gut microbiota in metabolic disorders is increasingly considered. Although microbiota is influenced by different factors, diet seems to be the major contributor of its diversity [1]. Both the type of diet and its caloric content are able to modify the relative proportion of gut microbes (increase of *Firmicutes* with parallel decrease of *Bacteroides*) and consequently their capability of harvesting energy from food [2,3].

The “energy harvest” hypothesis refers to the body’s ability to extract energy from resistant starch or dietary fiber that remains indigestible in the small intestine, which is well observed in subjects affected by obesity [4,5]. The fermentation of these residues produces short-chain fatty acids (SCFAs), which are used for lipid or glucose synthesis [6]. In some studies, subjects with obesity showed higher SCFAs (first propionate, followed by butyrate, valerate, and acetate) in the feces than their leaner controls, without differences in the characterization of the main bacterial phyla [7,8]. 

A relationship between human gut microbiota and metabolic disease exists, but what has to be clarified is whether the change in intestinal microbiota occurs before the development of inflammation or vice versa [9]. In humans, some studies showed that obesity is associated with a reduced bacterial diversity and an altered representation of bacterial species [10,11]; while Kasai et al. showed that bacterial diversity was significantly greater in subjects with obesity compared with subjects without obesity [12]. A metanalysis published in 2014 failed to show changes in microbial diversity between obese and non-obese populations [13].

*Firmicutes* and *Bacteroidetes* represent the two predominant phyla in murine and human microbiota and an alteration in this ratio is implicated in many diseases. It was first reported by Ley et al. that an increase in *Firmicutes* and a decrease in *Bacteroidetes* is associated with obesity [14]. This was subsequently confirmed by Kasai et al., who analyzed the gut microbiota of obese and non-obese Japanese subjects [12]. Their results showed a significant reduction of the number of *Bacteroidetes* and a higher *Firmicutes* to *Bacteroidetes* ratio in subjects with obesity compared to subjects with normal body weight. On the contrary, Schwiertz et al. reported a lower ratio of *Firmicutes* to *Bacteroidetes* in adults affected by overweight or obesity compared with individuals without weight problems, while Duncan et al. found no differences between *Firmicutes* and *Bacteroidetes* in subjects with different BMI [8,15]. Other studies showed a different pattern, characterized by a reduction of *Bacteroidetes* in individuals with obesity without differences in *Firmicutes* [10,16].

Angelakis et al. analyzed the duodenal microbial population in obese and non-obese subjects. They found that the phylum taxonomic profile was similar when subjects with obesity and the control group were compared, with small differences for *Firmicutes* (62% in the control group vs. 67% in the group with obesity) and *Proteobacteria* (9.5% in the control group vs. 4% in the obesity group) [17]. Unlike what is observed in the distal gut microbiota, *Bacteroidetes* were almost completely absent in the duodenum: This is probably due to a limited availability of mucin as a carbon source for *Bacteroidetes* [17].

In a study investigating the correlation between bacterial concentration and BMI, it was observed that the fecal concentration of *Lactobacillus reuteri* was positively correlated with BMI, while *Bifidobacterium animalis* and *Methanobrevibacter smithii* were negatively associated with BMI [18]. The gut microbiota associated with human obesity is depleted in *M. smithii* [19].

Since the discovery of the link between gut microbiota and metabolic health, attention was focused on the possible use of ingredients like probiotics as a therapeutically active or preventive strategy in the management of metabolic disease. According to the definition of the Food and Agriculture Organization (FAO) and World Health Organization (WHO), probiotics are live microorganism which, when administered in adequate amounts, confer a health benefit on the host [20]. Gram-positive bacteria, *Lactobacillus* and *Bifidobacterium,* are the two most common genera. Probiotics seem to have beneficial effects on obesity and related metabolic disorders [21]. A meta-analysis reported that *Lactobacillus gasseri* and *Lactobacillus plantarum* have positive effects on weight loss, while other species (*L. acidophilus*, *L. ingluviei* and *L. fermentum*) are associated with weight gain [22].

The present review focuses on the effectiveness of the probiotics in the reduction of body weight in overweight and obese subjects with metabolic diseases—previous reviews done by Aoun, Darwish and Hamod 2020 [23], and by Ballini et al., 2020 [24] indicated that probiotics modify the secretion of hormones, neurotransmitters, and inflammatory factors, thus preventing food intake triggers that lead to weight gain. The novelty of this review as compared to others is that the outcomes are related to the changes of anthropometric measures (weight, body mass Index (BMI), waist circumference (WC), and hip circumference (HC) after administration of various probiotics in overweight or obese subjects with metabolic diseases.

Given this background, the aim of this systematic review and meta-analysis is to evaluate the efficacy of probiotics as a potential treatment option to reduce body weight and ameliorate other anthropometric measures in overweight and obese patients with metabolic related diseases.

## 2. Materials and Methods 

The present systematic review was conducted in accordance with the Preferred Reporting Items for Systematic Review and Meta-Analysis (PRISMA) statement. The process of reporting was carried out as follows: (1) Formulation of working of research question stating that “is probiotic supplementation useful for the management of body weight and other anthropometric measures in adults affected by overweight and obesity with metabolic related diseases?”, (2) definition of participants: Adult women and men affected by overweight and obesity, (3) search strategy for identification of relevant intervention studies that include the effect of probiotic supplements on metabolic disease, and (4) analysis of data through the systematic review and meta-analysis.

### 2.1. Search Strategy

Articles that were written in the English language and published over the course of the last 10 years (2009–2019) were identified by searching PubMed and Scopus [25,26]. The search strategy was based on the following search terms: “probiotics” AND “obesity” AND “weight loss” AND “microbiota” OR “gut microbiota” AND “weight” AND “BMI” OR “WC” AND “HC”.

### 2.2. Study Eligibility

Eligible studies were required to report baseline and follow-up values, the mean change (∆-change) and relative standard deviation from baseline, and/or the mean difference among intervention groups vs. control group, concerning body weight or BMI and in addition other anthropometric measures, such as waist circumference (WC) and hip circumference (HC).

### 2.3. Data analysis and Presentation of Results

Randomized clinical trials investigating the effectiveness of the administration of different probiotic strains on body composition outcomes (especially weight loss) were included. For each study, the following data were specified: First author and the year of publication, the study design, the setting, the inclusion criteria, the number, and age of participants enrolled in the study, the intervention of the control and experimental group/s, the duration of the intervention and change of body measures observed in each group.

## 3. Results

A synthesis of the 20 published studies with 1411 patients is presented in Table 1. 

The table summarizes the studies that have evaluated as outcomes the changes of one or more anthropometric parameters (body weight, BMI, WC, and HC), after the administration of probiotics as supplements or food with a comparison between intervention and placebo treatment. In our analysis, we considered different study populations, including individuals with diabetes, obesity, non-alcoholic fatty liver disease (NAFLD), non-alcoholic steatohepatitis (NASH) metabolic syndrome, or altered lipid profile. 

### 3.1. Overweight and Obesity

Kadooka et al. were the first who evaluated the effect of the probiotic L. gasseri (LG2055) in overweight adults. They reported that subjects who had consumed fermented milk containing L. gasseri 2055 at a total dose of 10^11^ cfu/day showed a 4.6% reduction in visceral fat area after 12 weeks of treatment, which was significantly different from the placebo group [27]. Furthermore, the intervention group showed significant decreases in body weight, BMI, WC, HC, and waist-to-hip ratio at both weeks 8 and 12 of treatment, as compared with the control group. Three years later, the same study group confirmed these results, using a fermented milk containing a lower dose of LG2055. BMI, WC, HC, body fat mass, and visceral fat areas, in both 10^7^ and 10^6^ dose groups decreased significantly at weeks 8 and 12 from baseline [28]. 

On the other hand, another study showed no statistically significant changes of anthropometric parameters between subjects with hypertriaciglycerolemia, after taking fermented milk with or without LG2055. However, the active fermented milk appeared able to reduce postprandial and fasting serum non-esterified fatty acid levels, two important components of the risk for obesity and type 2 diabetes mellitus [29]. Jung et al. examined the efficacy of a treatment with L. gasseri BNR17 in adults affected by obesity and overweight [30]. The 12-week intervention revealed a slight reduction in body weight, WC, and HC. 

In contrast, no significant change in body composition was observed by Lee et al. in obese patients when the probiotics and placebo groups were compared [31]. In addition, Sanchez et al., showed that the administration of L. rhamnosus during the energy-restriction period (from week 1 to 12) did not significantly decrease the body weight or fat mass in a population of male and female obese patients [32]. The probiotic-treated group did, however, lose more fat mass than the placebo group at the end of the maintenance phase (from week 12 to 24). The analysis of the sex-specific results revealed significantly higher body weight and fat mass losses in women but not in men.

Another study investigated the changes in anthropometric parameters, in subjects affected by obesity or overweight, after the administration of probiotic yogurt (containing L. acidophilus La5, Bifidobacterium BB12, and L. casei DN001) combined or not with a low-calorie diet. The results showed a reduction in WC when probiotics were associated with a dietary restriction [33]. A recent study reported that the supplementation of a probiotic mix reduced abdominal adiposity and increased antioxidant enzyme activity in a more effective way when compared with an isolated dietary intervention. Participants taking the probiotic mix had a greater decrease in WC (−5.47% vs. −3.40%, *p* = 0.03), waist–height ratio (−5.00% vs. −3.27%, *p* = 0.02), and conicity index (−4.09% vs. −2.43%, *p* = 0.03) than the group receiving only the dietary intervention [34].

In addition, probiotics may improve weight loss after bariatric surgery. The results from the study of Woodard et al. suggested the use of a daily probiotic for all patients undergoing roux-en-Y gastric bypass, in order to reduce postoperative morbidity and maximize the weight loss [35].

### 3.2. Type 2 Diabetes Mellitus

Various studies have evaluated the effects of probiotics in overweight and obese patients affected by diabetes. They considered different kinds of probiotic supplementation, from multispecies probiotic supplements (with various strains of *Lactobacillus*, *Bifidobacterium,* and *Streptococcus*), to symbiotic food with *Lactobacillus sporogenes* and inulin. Comparing the anthropometric measures at baseline and after intervention, these studies failed to find a significant difference in weight and BMI between the two groups [36,37,38,39]. At the moment, other ongoing studies are evaluating the effects of probiotic supplementation in overweight and obese patients with prediabetes and diabetes [40,41].

### 3.3. NASH

A significant reduction (*p* < 0.001) of BMI was observed by Shavakhi et al. in overweight and obese patients affected by NASH with excess body weight, when treated with a combination of metformin and probiotics (different strains of *Lactobacillus* and *Bifidobacterium*) instead of metformin alone [42]. Also in children with NASH the administration of VLS#3 (a mixture of eight probiotic strains: *S. thermophilus*, bifidobacteria [*B. breve*, *B. infantis*, *B. longum*], *L. acidophilus*, *L. plantarum*, *L. paracasei*, and *L. delbrueckii* subsp. *bulgaricus*) significantly decreased (*p* < 0.001) the BMI during a four-month supplementation period, with respect to the placebo group [43]. Moreover, Nabavi et al. showed that body weight and BMI decreased in a significant manner in patients affected by obesity or overweight and NAFLD receiving a probiotic yogurt (4.42 × 10^6^ of *L. acidophilus* La5 and 3.85 × 10^6^ cfu/g of *B. lactis* Bb12), when compared with conventional yogurt, after an eight-week intervention [44].

The same bacterial strains, La5 and Bb12, were administered in a female population to assess their effects on the lipid profile. The participants were divided into three groups and were instructed to consume a daily dose of 300 g of probiotic yogurt, containing 3.9 × 10^7^ of both Bb12 and La5 or 300 g of conventional yogurt or consume any fermented and probiotic products. The authors reported mainly neutral effects of yogurt consumption on the lipid profile [45].

### 3.4. Metabolic Syndrome

Chang et al. reported benefits after the daily consumption of 300 mL of functional yogurt NY-YP901 consisting of several probiotics for eight weeks, on metabolic syndrome traits [46]. In particular, this kind of yogurt was associated with decreased low-density lipoprotein (LDL) cholesterol, body weight, and BMI following a daily consumption for eight weeks. Although there was no significant effect on the parameters of metabolic syndrome such as blood pressure, fasting blood glucose, triglycerides, and high-density lipoprotein (HDL), the decreases in LDL-cholesterol and body weight were expected to favor the decrease of cardiovascular risk [46]. In contrast, a study that aimed to investigate the effect of *L. casei* Shirota on gut permeability in patients with metabolic syndrome did not find any effect of this probiotic administration on BMI and WC [47].

The administration of 50 g/day of cheese containing *L. plantarum* TENSIA®, in subjects with hypertension, showed a reduction of BMI and blood pressure, that is, symptoms involved in metabolic syndrome. In these patients, a significant decrease in body weight was also observed when the intervention group was compared with controls (−5.7 vs. −4.4 kg, *p* = 0.083) [48]. 

### 3.5. Meta-Analyzed Data

The meta-analyzed mean differences for random effects (MD) showed no significant decrease in body weight after probiotic supplementation in patients with type 2 diabetes mellitus (Figure 1). 

In the 17 studies [27,28,29,30,31,32,33,34,35,36,37,38,39,40,41,42,43,44,45,46,47,48], with a total of 1057 subjects (536 in the intervention group and 521 in the control group), the overall effects showed that the treatment with probiotics did not significantly change the body weight (−0.26[−0.75, 0.23], *p* = 0.30) in the considered studies. τ^2^ (estimate of the between-studies variance in random-effect meta-analysis) = 0.94, *χ*^2^ = 556.40, df = 16 (*p* < 0.00001). *I^2^* (statistically describing the percentage of variation across studies that is due to heterogeneity) = 100%.

Figure 2 describes the meta-analyzed mean difference for random effects showing a significant decrease in BMI for the consumption of probiotic supplements. In a total of 18 studies, with a total of 1123 subjects (579 in the intervention group and 544 in the control group), the test of overall effects indicates that the treatment effect was significantly different (−0.73[−1.31, −0.16], *p* = 0.01) between the considered studies. τ^2^ (estimate of the between-studies variance in random-effect meta-analysis) = 1.36, *χ*^2^ = 3431.35, df = 15 (*p* < 0.00001). 

For WC (Figure 3) and HC (Figure 4), the meta-analyzed difference for random effects (MD) showed a significant decrease. For WC, 9 studies were included [27,28,29,30,31,32,33,34,35,36], with a total of 641 subjects (299 *n* the intervention group and 322 in the control group). Only 5 studies on HC were included [27,28,29,30,33], with a total of 407 subjects (190 in the intervention group and 217 in the control group). The test of overall effects for WC indicates that the treatment effect was significantly different (−0.71[−1.24, −0.19], *p* = 0.008) between the considered studies. τ^2^ (estimate of the between-studies variance in random-effect meta-analysis) = 0.53, *χ*^2^ = 221.93, df = 8 (*p* < 0.00001). *I^2^* (statistically describing the percentage of variation across studies that is due to heterogeneity) = 100%. Similarly, the test of overall effects for HC indicates that the treatment effect was significantly different (−0.73[−1.16, −0.30], *p* = 0.00008) between the considered studies. τ^2^ (estimate of the between-studies variance in random-effect meta-analysis) = 0.19, *χ*^2^ = 80.46, df = 4 (*p* < 0.00001). 

The risk of bias was also evaluated for the 20 studies: It was considered as high and low risk according to seven different criteria. Green indicates the risk of bias to be low, while red indicates the risk of bias to be high. 

## 4. Discussion

The results of this meta-analysis show that probiotic supplementation significantly decreases the BMI (−0.73 kg/m^2^ [−1.31, −0.16], *p* = 0.01), WC (−0.71 cm [−1.24, −0.19], *p* = 0.008), and HC (−0.73 cm [−1.16, −0.30], *p* = 0.0008), but not the body weight (−0.26 kg [−0.75, 0.23], *p* = 0.30) of adults of both sexes affected by overweight and obesity with metabolic related diseases. Probiotics seem to be mostly effective in NASH and metabolic syndrome patients.

According to this meta-analysis, the Lactobacillus (e.g., *L. Casei* strain Shirota (LAB13), *L. Gasseri, L. Rhamnosus, L. Plantarum*) and Bifidobacterium (e.g., *B. Infantis, B. Longum*, and *B. Breve B3*) show the most promising effects against obesity. A recent review shows promising in vivo and vitro effects of the same strains [49]. 

The definition of obesity was based on BMI and the intake of probiotics was followed by a significant decrease of this outcome. The decrease of WC and HC may also be linked to the decrease of BMI. The difference between the results obtained for BMI and body weight may be due to the lack of homogeneity among the studies included in this meta-analysis because some studies considered only BMI while other studies evaluated only body weight or both of them. Very important is the observed significant reduction of WC and HC because these parameters, particularly WC, are strictly related to cardiovascular risk [50]. One of the mechanisms involved in the reduction of BMI after probiotic intake is the regulation of gut microbiota. Obesity favors a change of the gut microbiota composition, which can affect the energy harvest from food, the secretory functions and the composition of adipose tissue, the metabolism of carbohydrates and lipids in the liver and could also influence the activity of specific centers in the brain [51]. The regulation of gut microbiota by means of probiotics is attained by enhancing the epithelial barrier integrity, increasing adhesion to intestinal mucosa (e.g., by increasing the amount of *Akkermansia muciniphila*), producing health-promoting and antimicrobial substances, excluding pathogenic microbes, and regulating the host immune system [51,52]. An increase in the amount of Firmicutes to Bacteroidetes leads to methylation of the obesity- and CVD-related genes and influences the activity of hormones affecting the metabolic function by increasing the ability to harvest energy [53].

Probiotics can play a significant role against obesity through species- and strain-specific mechanisms [49] *Lactobacillus reuteri* has shown that it can prevent the intestinal colonization of pathogenic microbes, by remodeling the commensal microbiota composition, by decreasing the production of pro-inflammatory cytokines, and by increasing the strength of the intestinal barrier [52]. *Lactobacullus paracasei* has shown in an animal model that it can decrease the fat storage by increasing the levels of angiopoietin-like 4 protein (ANGPTL4) [54]. *Lactobacillus gasseri* SBT2055 (LG2055) has shown that it can reduce lipid absorption and promote fecal fat excretion in humans [55].

Probiotics can influence effective on obese and diabetic patients, through positively influencing the lipid profile and insulin sensitivity—both mechanisms can have an ultimate positive effect on the body weight, BMI, WC, and HC [53]. Probiotics have shown that they decrease the total cholesterol, total triglycerides, and LDL levels, while they increase the level of HDL [53]. An increasing number of studies suggests that the oral and the intestinal microbiota may indirectly or directly influence cardiovascular risk. Besides diet, the other therapeutic and preventive route that could be traveled is that of microbiota modification, via the use of appropriate pro- and prebiotics [56].

In addition, probiotics also increase the production of short-chain fatty acids that eventually influence the appetite and energy homeostasis [51]. The enhanced production of short-chain fatty acids can affect inflammation resolution pathways in the mucosa [57]. A study done by Peng, Luying et al. concluded that butyrate enhances the intestinal barrier by regulating the assembly of tight junctions, mediated by the activation of AMPK [58]. An indirect mechanism of the anti-obesity activity of probiotics is through reversing the source of pro-inflammatory stimuli linked with low-grade endotoxemia and thus affecting the inflammatory response [57].

The administration of probiotics for the management of obesity may represent an attractive therapeutic strategy but, even though encouraging results emerged from experiments on rodents, the efficacy of probiotics in obese humans remains highly debatable [59].

The major limitations of this meta-analysis are due to the heterogeneity of the included studies. In particular, the age of patients showed a wide difference in the different studies and the same was for the duration on the intervention. The age of the participants ranged from 18 up to 75 years. The treatment duration also widely varied among the included studies, starting from 3 weeks up to 24 weeks. Both of these aspects negatively influence the data analysis and limit the understanding of the anti-obesity potential of probiotics.

Thus, in future research, it is essential to define several smaller age ranges while conducting clinical trials so that the effect of age becomes clearer. The treatment duration also widely varied between the studies, starting from 3 weeks up to 24 weeks. This wide range of duration contributes to limit the understanding of the anti-obesity potential of probiotics with respect to treatment duration.

This meta-analysis has various limitations based on the available scientific research, which is characterized by contradictory evidence. Part of the controversy is due to a lack of precise cost-effectiveness data and the lack of data on the correct dosage and type of probiotic that has to be supplemented. In addition, data on the correlation between specific claims and specific probiotics in obesity management are missing.

A second point of weakness of this meta-analysis is due to the absence of RCT in which the population sample is normalized for colonic content of bacteria. 

Conceivably, the obese patients have an increased Firmicutes/Bacteroidetes (F/B) ratio and might require different probiotic doses. In overweight/obese humans, in addition, the low fecal bacterial diversity is associated with more marked overall adiposity and dyslipidemia, impaired glucose homeostasis and higher low-grade inflammation [11].

Moreover, what constitutes a healthy microbiota is far from being established. For example, determining what constitutes a healthy microbiota and the variability found across populations is another important question mark. Recent studies raise questions about the widespread use of probiotics to impart general wellness [60].

There is huge variability of fermentable substrates that have bulk effects on bowel functions. day-by-day and within a day, since many studies have revealed only minor effects on overall microbiome composition and usually show only few species changing in population size [61].

Last, but not least, there is also no validated method to evaluate microbiota, most of which escapes current techniques, and in order to advance microbiome research to a more standardized and routine medical diagnostic procedure, it is essential to establish uniform standard operating procedures throughout laboratories and to initiate regular proficiency testing [62].

The “energy extraction” hypothesis should be interpreted with extreme caution. Daily energy output in feces is about 500 KJ and the microbiota produces SCFA, hence contributing to energy production rather than extraction. Note that rodents, especially the gnotobiotic models, are poor models of human microbiota behavior [61].

## 5. Conclusions

The results of this meta-analysis highlight a positive trend of probiotics supplementation on anthropometric measures of overweight and obese patients with related metabolic diseases. However, further research is needed before recommending the use of probiotics as a therapeutic strategy for these patients. The focus of the future research should be to evaluate the efficacy of different probiotic strains, the quantities to be administered, and the duration of the intervention.

## Figures and Tables

**Figure 1 nutrients-13-00666-f001:**
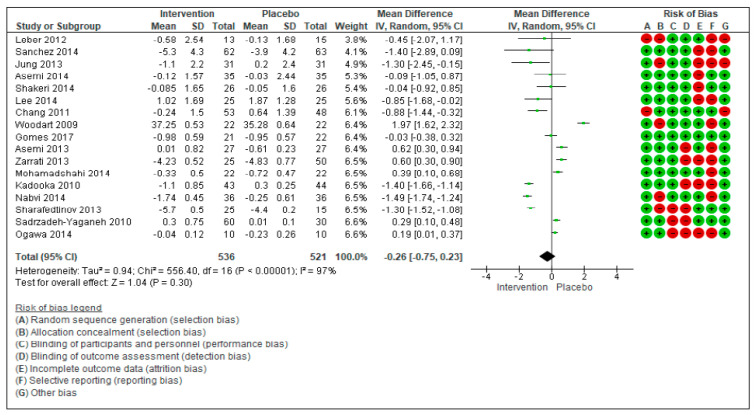
Forest plot for randomized controlled trials of probiotic supplementation included in body weight (kg) subgroup meta-analysis (*n* = 1057). The studies are listed by first author and year. IV = equation that can be estimated by inverse variance (linear, exponential). The square represents the measures of effect (i.e., an odd ratio) for each study; the area of each square is proportional to the study’s weight in the meta-analysis. Horizontal line represents the confidence interval (CI) at the 95% level. The diamond represents the meta-analyzed measure of effect; the lateral points of diamond indicate CIs for this estimate. The vertical line represents no effects; if the CI for an individual study overlaps with this line, the given level of confidence for the effect size does not differ from no effect for that study. Risk of bias indicates the level of high and low risk associated with the article. With green signal for low risk and red for high risk of bias.

**Figure 2 nutrients-13-00666-f002:**
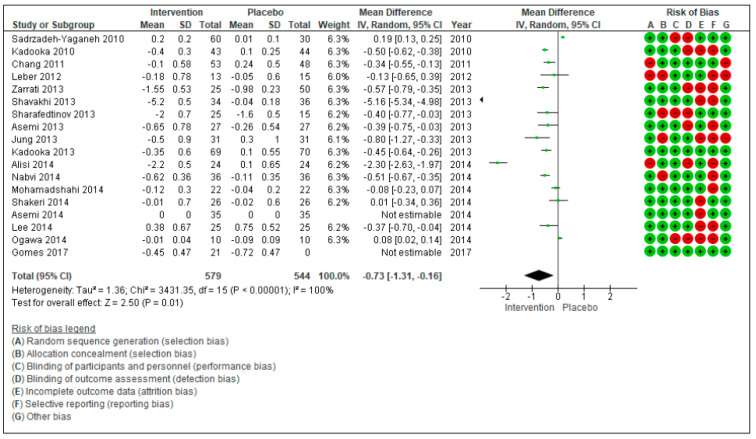
Forest plot for randomized controlled trials of probiotic supplementation included in body mass index (kg/m^2^) subgroup meta-analysis (*n* = 1123). The studies are listed by first author and year. IV = equation that can be estimated by inverse variance (linear, exponential). The square represents the measures of effect (i.e., an odd ratios) for each study; the area of each square is proportional to the study’s weight in the meta-analysis. Horizontal line represents the confidence interval (CI) at the 95% level. The diamond represents the meta-analyzed measure of effect; the lateral points of diamond indicate CIs for this estimate. The vertical line represents no effects; if the CI for an individual study overlaps with this line, the given level of confidence for the effect size does not differ from no effect for that study. Risk of bias indicates the level of high and low risk associated with the article, with green signal for low risk and red for high risk of bias.

**Figure 3 nutrients-13-00666-f003:**
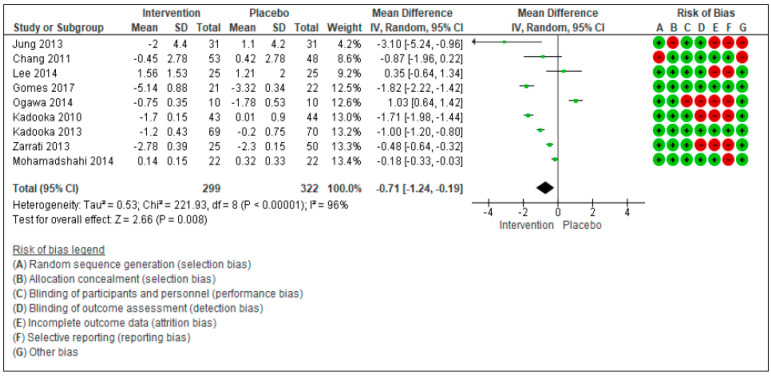
Forest plot for randomized controlled trials of probiotic supplementation included in waist circumference (cm) subgroup meta-analysis (*n* = 621). The studies are listed by first author and year. IV = equation that can be estimated by inverse variance (linear, exponential). The square represents the measures of effect (i.e., an odd ratios) for each study; the area of each square is proportional to the study’s weight in the meta-analysis. The horizontal line represents the confidence interval CI) at the 95% level. The diamond represents the meta-analyzed measure of effect; the lateral points of diamond indicate CIs for this estimate. The vertical line represents no effects; if the CI for an individual study overlaps with this line, the given level of confidence for the effect size does not differ from no effect for that study. Risk of bias indicate the level of high and low risk associated with the article, with green signal for low risk and red for high risk of bias.

**Figure 4 nutrients-13-00666-f004:**
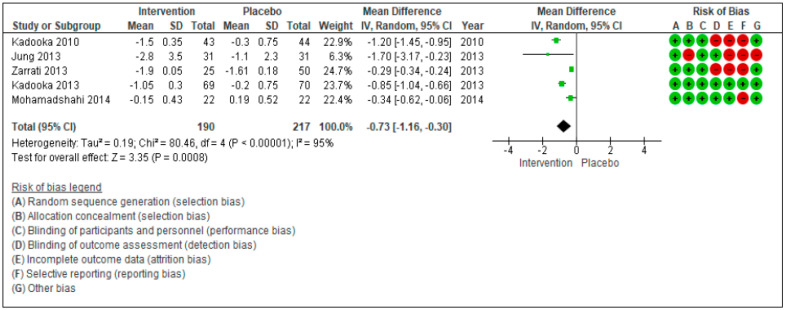
Forest plot for randomized controlled trials of probiotic supplementation included in hip circumference (cm) subgroup meta-analysis (*n* = 621). The studies are listed by first author and year. IV = equation that can be estimated by inverse variance (linear, exponential). The square represents the measures of effect (i.e., an odd ratios) for each study; the area of each square is proportional to the study’s weight in the meta-analysis. The horizontal line represents the confidence interval CI) at the 95% level. The diamond represents the meta-analyzed measure of effect; the lateral points of diamond indicate CIs for this estimate. The vertical line represents no effects; if the CI for an individual study overlaps with this line, the given level of confidence for the effect size does not differ from no effect for that study. Risk of bias indicate the level of high and low risk associated with the article, with green signal for low risk and red for high risk of bias.

**Table 1 nutrients-13-00666-t001:** Intervention studies on the effect of different probiotics on anthropometric parameters.

First Author, Year	Study Design	Participants (Age)	Intervention Group(s)	Placebo Group(s)	Duration	Changes in Intervention Group(s) ^a^	Changes in Control Group(s) ^a^
Gomes, 2017	Randomized controlled trial (RCT)	43 (20–59 years)	*n* = 21 Diet and 4 sachets/day: 1 × 10^9^ CFU of Lactobacillus acidophilus LA-14, L. casei LC-11, L. lactis LL-23, Bifidobacterium bifidum BB-06, and B. lactis BL-4	*n* = 22 diet	8 weeks	BW (kg): −0.98 BMI (kg/m^2^): −0.45 WC (cm): −5.14	BW (kg): −0.95 BMI (kg/m^2^): −0.72 WC: (cm) −3.32
Lee, 2014	RCT	50 (19–65 years)	*n* = 25 Twice/day Bofutsushosan, containing 18 components, 3 g per admnistration and priobiotic capsules (Duolac 7 included 5 billion viable of Streptococcus thermophiles, L. Plantarum, L. acidophilus, L. rhamnosus, B. Lactis, B. longum, and B. breve	*n* = 25 Twice/dayBTS (3 g per admnistration) and placebo capsules	8 weeks	BW (kg): 1.02 ± 1.69 BMI (kg/m^2^): 0.38 ± 0.67 WC (cm): 1.56 ± 1.53	BW (kg): 1.87 ± 1.28 BMI: 0.75 ± 0.52 WC (cm): 1.21 ± 2.00
Sanchez, 2014	RCT	125 (18–55 years)	*n* = 62 two capsules daily (6 × 10^8^cfu of L. rhamnosus CGMCC1.3724 (LPR))	*n* = 63 Two capsules daily	24 weeks	BW (kg): −5.3 ± 4.3	BW (kg): −3.9 ± 4.2
Zarrati, 2013	RCT	75 (20–50 years)	Group 1, -, *n* = 25: diet and 200 g/day of probiotic yogurt (PLCD), containing S. thermophiles and L. bulgaricus - enriched with the L. acidophilus LA5, L. casei DN001 and, B. lactis Bb12 (1 × 10^8^ cfu/g each strain)	Group 2, *n* = 25: diet and 200 g/die of regular yogurt (RLCD)- Group 3, *n* = 25: 200 g/day of probiotic yogurt without any diet (PWLCD)	8 weeks	PLCD: BW (kg): −4.23 BMI (kg/m^2^): −1.3 WC (cm): −2.78 HC (cm): −2	RLCD: BW (kg): −4.87 BMI (kg/m^2^): −1.9 WC (cm): −2.3 HC (cm): −3.18 PWLCD: BW (kg): −0.04 HC (cm): −0.03
Jung, 2013	RCT	62 (19–60 years)	*n* = 31 6 capsules/day composed of 10^10^ cfu of L. gasseri BNR17	*n* = 31 6 placebo capsules/day	12 weeks	BW (kg): (−1.1 ± 2.2) BMI (kg/m^2^): (−0.5 ± 0.9) WC (cm): (−2 ± 4.4) HC (cm): (−2.8 ± 3.5)	BW (cm): (0.2 ± 2.4) BMI (kg/m^2^): (0.3 ± 1.0) WC (cm): (1.1 ± 4.2) HC (cm): (−1.1 ± 2.3)
Sharafedtinov, 2013	RCT	40 (30–69 years)	*n* = 25 50 g/day of probiotic product (semi-hard cheese) containing L. plantarum TENSIA, added in amounts of 1.5 × 10^11^	*n* = 15 50 g/day of cheese without probiotics	3 weeks	BW (kg): −5.7 BMI (kg/m^2^): −2	BW (kg): −4.4 BMI (kg/m^2^): −2.3
Kadooka, 2013	RCT	210 (35–60 years)	*n* = 69 Fermented milk (FM) containing 10^7^ cfu LG2055/g *n* = 71 FM containing 10^6^ cfu LG2055/g	*n* = 70 Control FM containing 0 cfu LG2055/g	12 weeks	10^7^ dose BMI (kg/m^2^): (−0.3) WC (cm): (−1.3) HC (cm): (−1.2) 10^6^ dose BMI (kg/m^2^): (−0.4) WC (cm): (−1.1) HC (cm): (−0.9)	BMI (kg/m^2^): (0.1) WC (cm): (−0.1) HC (cm): (−0.2)
Kadooka, 2010	RCT	87 (33–63 years)	*n* = 43 200 g daily of FM with L. gasseri SBT2055 (LG2055), 5 × 10^10^ cfu/100 g of FM	*n* = 44 200 g (2 portions of 100 g each) daily of FM without LG2055	12 weeks	BW (kg): −1.1 BMI (kg/m^2^): −0.4 WC (cm): −1.7 HC (cm): −1.5	BW (kg): 0.3 BMI (kg/m^2^): 0.1 HC (kg): −0.3
Woodard, 2009	RCT	44 (median age of treated group was 48.6 years, of placebo group was 41.2)	*n* = 22 1 pill/day of Puritan’s Pride®, probiotic supplement containing 2.4 billion live cells of Lactobacillus species.	*n* = 22 placebo	24 weeks	Weight loss % (6 weeks postoperative): 29.90	Weight loss % (6 weeks postoperative):25.50
Asemi, 2014	RCT	70 (35–70 years)	*n* = 35 3 times/day of synbiotic food with L sporogenes (1 × 10^7^ cfu) and 0.04 inulin as prebiotic. Then they received 27 × 10^7^ cfu L. sporogenes and 1.08 g of inulin each day	*n* = 35 Control food: the same substance without probiotic bacteria and prebiotic inulin	6 weeks	BW (kg): (−0.12 ± 1.57) BMI (kg/m^2^): (−0.05 ± 0.62)	BW (kg): (−0.03 ± 2.44) BMI (kg/m^2^): (−0.02 ± 1)
Asemi, 2013	RCT	54 (35–70 years)	*n* = 27 The probiotic supplement has L. acidophilus (2 × 10^9^ cfu), L. casei (7 × 10^9^ cfu), L.rhamnosus (1.5 × 10^9^ cfu), L.bulgaricus (2 × 10^8^ cfu), B. breve (2 × 10^10^ cfu), B.longum (7 × 10^9^ cfu), S. thermophilus (1.5 × 10^9^ cfu) and 100 mg fructo-oligosaccharides	*n* = 27 Placebo: the same substance without bacteria	8 weeks	BMI (kg/m^2^): −0.65	BW (kg): −0.61 BMI (kg/m^2^): −0.26
Shakeri, 2014	RCT	78 (35–70 years)	*n* = 26 The synbiotic bread contained probiotic L. sporogenes (1 × 10^8^ cfu) and 0.07 g inulin as prebiotic per 1 g. *n* = 26 The probiotic bread contained L. sporogenes (1 × 10^8^ cfu) per 1 g.	*n* = 26 Control bread: the same substance without probiotic bacteria and prebiotic inulin	8 weeks	Synbiotic bread: BW (kg): (0.03 ± 1.9) BMI (kg/m^2^): (0.02 ± 0.8) Probiotic bread: BW (kg): (-0.2 ± 1.4) BMI (kg/m^2^): (−0.04 ± 0.6)	Control bread: BW (kg): (−0.05 ± 1.6) BMI (kg/m^2^): (−0.02 ± 0.6)
Mohamadshahi, 2014	RCT	44 (18–70 years)	*n* = 22 300 g/day of probiotic yogurt (L. delbrueckii subsp. bulgaricus and S. thermophilus + 3.7×10^6^ cfu/g of both B. animalis subsp. lactis Bb12 and L. acidophilus strain La5	*n* = 22 300 g/day of conventional yogurt	8 weeks	BW (kg): −0.33 BMI (kg/m^2^): −0.12 WC (cm): 0.5 HC (cm): −0.15	BW (kg): −0.72 BMI (kg/m^2^): −0.04 WC (cm): 0.34 HC (cm): 0.19
Nabavi, 2014	RCT	72 (23–63 years)	*n* = 36 300 g/day of probiotic yogurt containing L. acidophilus La5 (4.42 × 10^6^ cfu/g) and B. lactis Bb12 (3.85 × 10^6^ cfu/g)	*n* = 36 300 g/day of conventional yogurt	8 weeks	BW (kg): −1.74 BMI (kg/m^2^): −0.68	BW (kg): −0.25 BMI (kg/m^2^): −0.11
Alisi, 2014	RCT	48 (median age of treated group was 11 years, of placebo group was 10 years)	*n* = 24 Probiotic VLS#3, 1 sachet/day <10 years old or 2 sachet/day >10 years old	*n* = 24 Placebo, 1 sachet/day <10 years old or 2 sachet/day >10 years old	16 weeks	BMI (kg/m^2^): −2.2	BMI (kg/m^2^): 0.1
Shavakhi, 2013	RCT	70 (18–75 years)	*n* = 34 Two tablets/day of metformin 500 mg + two tablets/day of Protexin (L. acidophilus 1 × 10^8^ CFU, L. casei 5 × 10^8^ CFU, L. rhamnosus 7.5 × 10^7^ CFU, L. bulgaricus 1.5 × 10^8^ CFU, B. breve 5 × 10^7^ CFU, B. longum 2.5 × 10^7^ CFU, S. thermophilus 5 × 10^7^ CFU, fructooligosaccharides 350 mg)	*n* = 36 Two tablets/die of metformin 500 mg + two placebo tablets (120 mg of starch)/day	24 weeks	BMI (kg/m^2^): −5.2	BMI (kg/m^2^): −0.44
Leber, 2012	RT	28 (24–66 years)	*n* = 13 3 bottles/day (65 ml) containing L. casei Shirota at a concentration of 10^8^/ml (3 × 6.5 × 10^9^ cfu L. casei Shirota)	*n* = 15 not received the product and served as a control group (standard).	12 weeks	BW (kg): (−0.58 ± 2.54) BMI (kg/m^2^): (−0.18 ± 0.78)	BW (kg): (−0.13 ± 1.68) BMI (kg/m^2^): (−0.05 ± 0.60)
Chang, 2011	RCT	101 (20–65 years)	*n* = 53 Functional yogurt containing S. thermophilus ≥3 × 10^9^c.f.u./g, L. acidophilus ≥3 × 10^9^c.f.u./g, B. infantis ≥1 × 10^10^c.f.u./g and functional ingredients	*n* = 48 The control yogurts contained the same ingredients of S. thermophilus, L. acidophilus, B. infantis except functional ingredients	8 weeks	BW (kg): (−0.24 ± 1.50) BMI (kg/m^2^): (−0.10 ± 0.58) WC (cm): (−0.45 ± 2.78)	BW (kg): ( + 0.64 ± 1.39) BMI (kg/m^2^): ( + 0.24 ± 0.50) WC (cm): ( + 0.42 ± 2.78)
Ogawa, 2014	Single-blind, CT	30 (27–69 years)	*n* = 15 200 g (2 portions of 100 g each) daily of FM with L. gasseri SBT2055 (LG2055) The viable cell count of LG2055 waproximately 5 × 10^10^ cfu/100g of FM on the initial day	*n* = 15 200 g (2 portions of 100 g each) daily of control FM without LG2055 L. gasseri SBT2055 (LG2055)	Control FM for 4 weeks; 4 weeks of washout period, active FM for 4 weeks	BW (kg): (−0.04 ± 0.12) BMI (kg/m^2^): (−0.01 ± 0.04) WC (cm): (−0.75 ± 0.35)	BW (kg): (−0.23 ± 0.26) BMI (kg/m^2^): (−0.09 ± 0.09) WC (cm): (−1.78 ± 0.53)
Sadrzadeh-Yaganeh, 2010	RCT	90 (19–49 years)	Group 1: *n* = 30 consumed daily 300 g probiotic yogurt containing L acidophilus La5 and B. lactis Bb12 (3.9 × 10^7^ of both Bb12 and La5) Group 2: *n* = 30 consumed daily 300 g conventional yogurt	Group 3: *n* = 30 did not consume any fermented and probiotic products	6 weeks	Group 1 BW (kg): 0.2 Group 2 BW (kg): 0.4 BMI (kg/m^2^): 0.2	Group 3 No changes

^as^ Changes expressed as: (∆ change) ± SD where data are available. Abbreviations: CFU, colony forming unit; BW, body weight; BMI, body mass index; WC, waist circumference; HC, hip circumference; PLCD, probiotic yogurt with low calorie diet; RLCD, regular yogurt with low calorie diet; PWLCD, probiotic yogurt without low calorie diet; FM, fermented milk.

## Data Availability

Not applicable.

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
