# Peer review of "Is Probiotic Supplementation Useful for the Management of Body Weight and Other Anthropometric Measures in Adults Affected by Overweight and Obesity with Metabolic Related Diseases? A Systematic Review and Meta-Analysis"

_nutrients, 2021, doi:10.3390/nu13020666_

Round 1

Reviewer 1 Report

 My concerns:

  1. Discussion: authors need to state the novelty of this review compared to other review articles in this fied.
  2. Discussion, lane 351: this study  needs to be added - butyrate enhances the intestinal barrier by facilitating tight junction assembly (J Nutr. 2009 Sep; 139(9): 1619–1625).

Author Response

QUESTION

  1. Discussion: authors need to state the novelty of this review compared to other review articles in this field.

(Answer.) The novelty of this review has been declared (page 3, line 101)

QUESTION

  1. Discussion, lane 351: this study needs to be added - butyrate enhances the intestinal barrier by facilitating tight junction assembly (J Nutr. 2009 Sep; 139(9): 1619–1625).

(Ans.) The above-mentioned study outcome has been added to shows the effect of butyrate on intestinal barrier. (page 13, line 358)

Reviewer 2 Report

The systemic review based on meta-analysis gives extensive knowledge about the effect of probiotic supplements in the induction of weight loss. This review article is well designed and structured to explain the reported literature on the anti-obesity potential of probiotics on body weight.

Overall, the manuscript is written in a coordinated manner to provide clear information. However, there are few minor revisions to be made within the manuscript and also several typos throughout the main text to be rectified.

  1. There are several typo errors throughout the main text to be rectified. For example, In abstract line 27, there is a repetition of brackets ‘(HC))..’ In line 129, please check the word ‘group/s’.
  2. The spacing between words in few sentences is not constant. The authors should check and unify it. For example, In line 148, rectify the space after the word ‘adults .’
  3. A re-check of the grammatical errors in the manuscript will be better to improve the quality of the manuscript.

Author Response

  1. There are several typo errors throughout the main text to be rectified. For example, In abstract line 27, there is a repetition of brackets ‘(HC))..’ In line 129, please check the word ‘group/s’.
  2. The spacing between words in few sentences is not constant. The authors should check and unify it. For example, In line 148, rectify the space after the word ‘adults .’
  3. A re-check of the grammatical errors in the manuscript will be better to improve the quality of the manuscript.

(Ans.) A new evaluation of the text has been done to clarify all the grammatical mistakes.  All the errors have been corrected to the best of knowledge with the assistance of a professional native English proofreader.

Reviewer 3 Report

nutrients-1092543

This is an interesting review addressing a currently over-hyped topic, i.e. the use of probiotics (instead of dieting) to lose weight. Everyone wants to eat ad libitum and take the magic pill (from the useless resveratrol to the equally useless probiotics).

It is hard to believe that it took 12 authors to carry out this review. Better check ICMJE (e.g. project administration?).

The authors reach and confirm the same conclusions of PMID:32492487 (not discussed), i.e. that nothing works. That paper did not acknowledge some limitations of microbiota research, which must be discussed here. No one normalizes for colonic content of bacteria. Conceivably, obese patients have more microbes and might require different doses. Also, what constitutes a healthy microbiota is far from being established. Indeed, some authors suggest that giving probiotics does more harm than good PMID: 30698619. Note that there is huge variability in stool composition, day-by-day and within a day PMID: 32238220; PMID: 26246784. There is also no validated method to evaluate microbiota, most of which escapes current techniques PMID: 27052158.

The “energy extraction” hypothesis should be interpreted with extreme caution PMID: 32238220. Daily energy output in feces is about 500 KJ and the microbiota produces SCFA, hence contributing to energy production rather than extraction. Note that rodents, especially the gnotobiotic models, are poor models of human microbiota behavior PMID: 32238220

In summary, this review confirms current knowledge and there are plenty of references to be discussed by the large confederacy of authors.

Author Response

QUESTION

This is an interesting review addressing a currently over-hyped topic, i.e. the use of probiotics (instead of dieting) to lose weight. Everyone wants to eat ad libitum and take the magic pill (from the useless resveratrol to the equally useless probiotics).

ANSWER Obviously the magic pill to treat obesity does not exist. This paper focuses on the potential role of probiotics to favor microbiota changes that could help obesity treatment, given the available knowledge on the differences in microbiota when normal and overweight subjects are compared

QUESTION

It is hard to believe that it took 12 authors to carry out this review. Better check ICMJE (e.g. project administration?).

The authors reach and confirm the same conclusions of PMID:32492487 (not discussed), i.e. that nothing works.

ANSWER In the words of Federico Scarmozzino , Andrea Poli , and Francesco Visioli , (who wrote PMID:32492487 ):   1)  “an increasing number of studies suggests that the oral and the intestinal microbiota may indirectly or directly influence cardiovascular risk. “2) Beside diet,” the other therapeutic and preventive route that could be traveled is that of microbiota modification, via the use of appropriate pro- and prebiotics.” 

So they , similarly to what observed in our paper,  did not conclude their interesting paper  saying that nothing works, but that this is an interesting research area that could generate future new therapeutical approaches as soon as clear data will be available on microbiota in overweight and metabolic disorders as well as on the effects of specific probiotics will be available . We thank very much reviewer 3 for the suggestion to include some considerations derived from PMID:32492487 in the discussion of our paper

QUESTION

That paper did not acknowledge some limitations of microbiota research, which must be discussed here. No one normalizes for colonic content of bacteria. Conceivably, obese patients have more microbes and might require different doses. Also, what constitutes a healthy microbiota is far from being established. Indeed, some authors suggest that giving probiotics does more harm than good PMID: 30698619. Note that there is huge variability in stool composition, day-by-day and within a day PMID: 32238220PMID: 26246784. There is also no validated method to evaluate microbiota, most of which escapes current techniques PMID: 27052158. The “energy extraction” hypothesis should be interpreted with extreme caution PMID: 32238220. Daily energy output in feces is about 500 KJ and the microbiota produces SCFA, hence contributing to energy production rather than extraction. Note that rodents, especially the gnotobiotic models, are poor models of human microbiota behavior PMID: 32238220

In summary, this review confirms current knowledge and there are plenty of references to be discussed by the large confederacy of authors.

(Ans.) Thank you for your valuable guidance in order to underline various limitations of the microbiota research. We fully included the suggested topics and related references in the discussion section (page 13,14 line 377-419)

Round 2

Reviewer 3 Report

Good revision in which the many Authors confirm that nothing works. Maybe in the future.... who knows...

Author Response

We thank very much the reviewer for the stimulating comments and suggestions. We think that this topic is of relevant importance and doctors as well as patients are looking for positive results and indications for future therapeutic intervention with probiotics in overweight and obese patients. Our results show that up to now it is impossible to suggest such a treatment.